# Natural and Engineered Electron Transfer of Nitrogenase

**Wenyu Gu [1] and Ross D. Milton [2],\***

[1]   Department of Civil and Environmental Engineering, Stanford University, E250 James H. Clark Center, 318 Campus Drive, Stanford, CA 94305, USA

[2]   Department of Inorganic and Analytical Chemistry, University of Geneva, Sciences II, Quai Ernest-Ansermet 30, 1211 Geneva 4, Switzerland

\*   Correspondence: ross.milton@unige.ch

**Abstract:** As the only enzyme currently known to reduce dinitrogen ($N_2$) to ammonia ($NH_3$), nitrogenase is of significant interest for bio-inspired catalyst design and for new biotechnologies aiming to produce $NH_3$ from $N_2$. In order to reduce $N_2$, nitrogenase must also hydrolyze at least 16 equivalents of adenosine triphosphate (MgATP), representing the consumption of a significant quantity of energy available to biological systems. Here, we review natural and engineered electron transfer pathways to nitrogenase, including strategies to redirect or redistribute electron flow in vivo towards $NH_3$ production. Further, we also review strategies to artificially reduce nitrogenase in vitro, where MgATP hydrolysis is necessary for turnover, in addition to strategies that are capable of bypassing the requirement of MgATP hydrolysis to achieve MgATP-independent $N_2$ reduction.

**Keywords:** nitrogenase; ammonia; metalloenzyme; electron transfer; ferredoxin; flavodoxin; Fe protein; MoFe protein

---

## 1. Introduction to Nitrogenase

Ammonia ($NH_3$) is an important commodity for agricultural and chemical industries that is currently produced at over 150 million tons per year [1,2]. Currently, the majority of this $NH_3$ is produced from molecular hydrogen ($H_2$) and kinetically inert dinitrogen ($N_2$, bond dissociation enthalpy of +945 kJ mol$^{-1}$) by the Haber–Bosch process, which operates at a high temperature (~700 K) and a high pressure (~100 atm) in order to optimize $NH_3$ production [3]. These conditions, in combination with the production of $H_2$ (commonly by steam reforming of natural gas), result in the consumption of 1–2% global energy and the production of around 3% of global carbon dioxide ($CO_2$) emissions. Due to ever-increasing concerns and awareness of climate change, there is significant interest in the development of new catalysts for $N_2$ fixation. For instance, the development of a new (bio)catalytic system that operates under mild conditions could enable the decentralization of $NH_3$ production as a key strategy for improved environmental sustainability.

Select bacteria and archaea are able to produce an enzyme, nitrogenase, which can fix $N_2$ to $NH_3$ under mild (physiological) conditions [4]. Thus, nitrogenase is of interest to new biotechnologies and new bio-inspired $N_2$-fixing catalysts. Nitrogenase is a two-component metalloenzyme consisting of a reductase (iron or "Fe" protein) and a $N_2$-reducing protein (MoFe protein), where the name "MoFe" refers to the metals employed in its catalytic cofactor. There are two alternative nitrogenases that are dependent on V (VFe) and Fe only (FeFe), although the Mo-nitrogenase system from the soil bacterium *Azotobacter vinelandii* will serve as the model to outline nitrogenase's mechanism (Figure 1) [5].

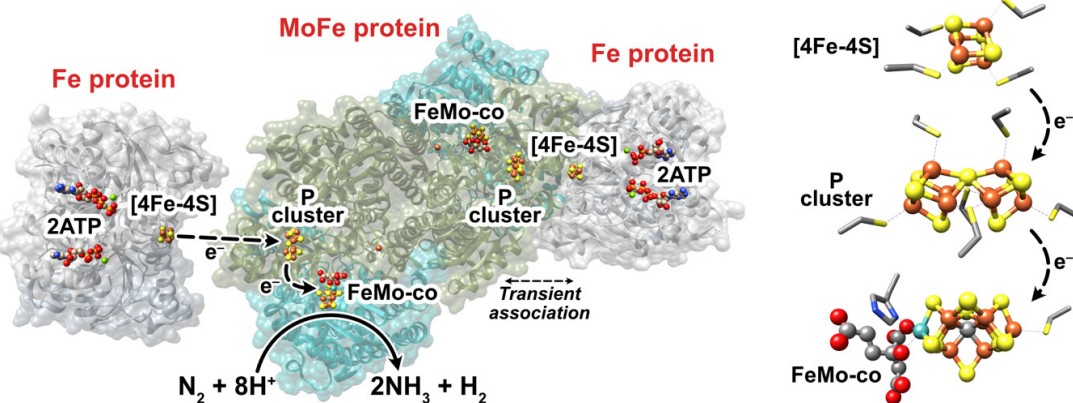

**Figure 1.** (**Left**) Representation of Mo-nitrogenase from *Azotobacter vinelandii*. The Fe protein is shown in gray and the MoFe protein is shown in cyan (NifD, α-subunit) and olive-green (NifK, ß-subunit). This representation was adapted from PDB:4WZA, which used non-hydrolyzable ATP analogues to form a tight Fe:MoFe protein complex. (**Right**) The FeS cofactors of Mo-dependent nitrogenase, where the coordinating residues are also shown. The homocitrate partner of the FeMo-co is shown on the left of the FeMo-co. Fe = rust, S = yellow, C = gray, N = blue, O = red, Mo = cyan, Mg = green.

The Fe protein of *A. vinelandii* is a homodimer of approximately 66 kDa in mass encoded by the *nifH* gene. Each dimer contains an adenosine triphosphate (MgATP) binding site, whereas a [4Fe-4S] iron-sulfur cluster is located between each monomer and coordinated by two cysteine (Cys) residues from each monomer [6]. The MoFe protein is a $\alpha_2\text{ß}_2$ tetramer of approximately 240 kDa encoded by the *nifD* (α subunit) and *nifK* (ß subunit) genes. Each αß dimeric half contains a [8Fe-7S] "P" cluster that bridges both subunits (coordinated by Cys residues from each subunit) and a [7Fe-9S-C-Mo-homocitrate] FeMo-cofactor ("FeMo-co") contained within the α subunit [7,8]. The FeMo-co, the cofactor at which $N_2$ is reduced, is coordinated by a Cys and a histidine (His) residue. Each αß half of the MoFe protein repeatedly transiently associates with a $1e^-$-reduced and ATP-bound Fe protein, during which a single electron is ultimately transferred from the $[4Fe-4S]^{1+}$ cluster to the FeMo-co *via* the P cluster [9]. The order of events that transpire during this transient association is debated, although electron transfer, 2MgATP hydrolysis (to adenosine diphosphate, 2MgADP), the release of two inorganic phosphate ($P_i$) equivalents, and Fe protein dissociation occur. Oxidized Fe protein is subsequently reduced by flavodoxin and/or ferredoxin in vivo or commonly dithionite (DT) in vitro, thereby restarting the "Fe protein cycle" [10,11]. The reduction potential ($E^{o'}$) of the $[4Fe-4S]^{2+/1+}$ couple of the Fe protein is approximately −0.30 V vs. the standard hydrogen electrode (SHE, see [12,13]), free of MgATP or MgADP [9,14]. Upon the association of MgATP, the $E^{o'}$ of $[4Fe-4S]^{2+/1+}$ is modulated to −0.43 V vs. SHE, which decreases further to −0.62 V when the Fe protein associates with the MoFe protein making it a more potent electron donor [14,15]. Perhaps the two most-relevant redox couples of the P cluster ($P^{N/1+}$ and $P^{1+/2+}$) both have $E^{o'}$'s of approximately −0.31 V vs. SHE. Further, the $E^{o'}$ of the as-isolated FeMo-co ($M^N$) and its one-electron oxidation production ($M^{OX}$) is approximately −0.04 V vs. SHE, while a second lower-potential redox couple ($M^{RED}$) is also thought to be relevant to nitrogenase's catalytic cycle, although its $E^{o'}$ has not been accurately determined [9].

Electron transfer from the Fe protein is currently thought to be coupled to the rate-limiting step of $P_i$ release (25–27 $s^{-1}$), suggesting electron transfer takes place with a rate constant of around 13 $s^{-1}$ [16]. Along with a study by Duval et al. in 2013, this suggests that electron transfer from the $[4Fe-4S]^{1+}$ cluster to the P cluster takes place prior to MgATP hydrolysis [17]. Finally, a deficit spending model has been proposed by which an electron is transferred from the P cluster to the FeMo-co, prior to the P cluster being reduced by the Fe protein's $[4Fe-4S]^{1+}$ cluster [18,19]. Nevertheless, the proposal that electron transfer from the Fe protein's [4Fe-4S] cluster takes place prior to the hydrolysis of MgATP suggests that artificial electron transfer to the P cluster (independent of the Fe protein) is

possible. Thus significant interest has developed concerning the MgATP-independent fixation of $N_2$ by nitrogenase, given that the hydrolysis of each MgATP accounts for around $-50$ kJ mol$^{-1}$ in vivo [20].

The reduction of $N_2$ to $NH_3$ requires 6e$^-$, although optimal $N_2$ fixation by nitrogenase occurs after eight transient association events of the Fe protein (and the transfer of 8e$^-$):

$$N_2 + 8e^- + 8H^+ + 16MgATP \rightarrow 2NH_3 + H_2 + 16MgADP + 16P_i \tag{1}$$

Notably, the fixation of each $N_2$ also results in the evolution of at least one equivalent of $H_2$. Lowe and Thorneley developed an early model to describe the observation of $H_2$ formation, which has since developed into a model that highlights the pivotal nature of a 4e$^-$-reduced FeMo-co known as the $E_4$ state [8,10,21–23]. By this model, the resting FeMo-co in its $E_0$ state accumulates individual electrons and protons during each Fe protein association event in order to reach the $E_4$ state at which $N_2$ binds and undergoes subsequent reduction. These electrons are stored as metal-hydrides (M-H) on the Fe centers. Prior to the binding of $N_2$, the $E_{2-4}$ states can unproductively evolve $H_2$ by M-H protonation, thereby "dropping" by $E_{x-2}$ states [21,22]. In contrast, the productive evolution of $H_2$ is thought to occur in order to accommodate $N_2$ binding and its partial reduction, by the reductive elimination (*re*) of $H_2$ from the FeMo-co $E_4$ state [24]. At this stage, $H_2$ can also undergo oxidative addition (*oa*) to the FeMo-co and displace $N_2$, which is also considered to be unfavorable [22]. Four additional Fe protein association events (transferring 4e$^-$ and hydrolyzing an additional 8MgATP) then lead to the production of 2$NH_3$. This serves as a suitable model to explain the production of one equivalent of $H_2$ for each $N_2$ reduced [21,22]. Thus, $H_2$ can be produced by unproductive or productive pathways, with increased $H_2$ evolution and MgATP hydrolysis resulting from a combination of both. Questions therefore remain surrounding the reversibility or catalytic bias of the $E_4$ state, given that the alternative nitrogenases also appear to follow the same *re* mechanism while appearing to be less-efficient at $N_2$ fixation [5,25]. A "just-in-time" mechanism could serve as a useful model to justify the rate-limiting nature of electron transfer from the Fe protein (~13 s$^{-1}$) such that unproductive $H_2$ formation and the *oa* of $H_2$ is minimized [23]. Finally, research has also questioned the suitability of the proposed *re/oa* model for $N_2$ fixation by nitrogenase [26]. Density functional theory calculations were employed to calculate the partial charges of the FeMo-co during key turnover states, where a *re* mechanism was not supported and the possible involvement of hydrogen atoms (including hydrogen atom transfer, formation, and elimination steps) was highlighted.

In summary, it is clear that electron transfer within the nitrogenase complex, and by extension, to the nitrogenase complex, is of high importance to biological $N_2$ fixation as well as new biotechnologies and bio-inspired $N_2$ fixation systems. Thus, this article reviews natural and engineered electron transfer to nitrogenase, in the contexts of both in vivo and in vitro catalysis. Specifically, we discuss the nature and delivery of electrons to nitrogenase in vivo. We also review approaches to transfer electrons to nitrogenase in vitro, either through the Fe protein or independent of the Fe-protein for MgATP-decoupled catalysis by the MoFe protein.

## 2. In Vivo Electron Transfer to Nitrogenase

### 2.1. Electron Transfer from Ferredoxin and Flavodoxin

In vivo reduction of Fe protein requires low-potential electrons provided by the electron-transferring proteins flavodoxin and ferredoxin [27,28]. In vitro studies of nitrogenase typically utilize dithionite (DT, $E^{o'} \approx -0.66$ V vs. SHE) or Ti(III) citrate ($E^{o'}_{Ti(III)/(IV)} = -0.8$ V vs. SHE) as electron donors due to their ease of preparation and use [29–31]. However, recent studies have reevaluated electron transfer to the Fe protein with flavodoxin, which is crucial for a mechanistic understanding of the Fe protein cycle and of nitrogenase activity in physiological conditions [16]. It is also important to consider physiological electron donors when developing biological systems to maximize electron transfer to nitrogenase.

Flavodoxin (Fld) is a monomeric electron-transfer protein carrying a non-covalently bound flavin mononucleotide (FMN) as a redox center. It consists of a central five-stranded parallel β-sheet flanked

on either side by α-helices. Based on the presence of a ~20-residue loop splitting the fifth β-strand, Flds involved in electron transfer to nitrogenase are classified as long-chain flavodoxins [32,33]. The cofactor FMN has three pertinent redox states: oxidized quinone (Ox), semiquinone (SQ), and hydroquinone (HQ), where $FMN^{Ox/SQ}$ and $FMN^{SQ/HQ}$ couples are $1e^-$ redox reactions. Although not overly abundant in aqueous solution, $FMN^{SQ}$ is stabilized when bound to flavodoxin; further, crossed-potentials of flavins can be stabilized in proteins and can support a process named flavin-based electron bifurcation (reviewed in [34–36]). $E^{o'}$'s of Fld II from *A. vinelandii* range from −0.25 to −0.1 V (vs. SHE) for the $Fld^{Ox}/Fld^{SQ}$ couple and from −0.5 to −0.4 V (vs. SHE) for the $Fld^{SQ}/Fld^{HQ}$ couple [32]. The latter is one of the lowest reported potentials in the flavodoxin family [37]. Because the $E^{o'}$ of the Fe protein is <−0.3 V (vs. SHE), the $Fld^{HQ}$ state is expected to be the functional electron donor [27,38]. In *A. vinelandii*, Fld ($E^{o'}_{SQ/HQ}$ = −0.46 V vs. SHE) is encoded by the gene *nifF*.

Ferredoxin (Fdx) is an FeS cluster-containing protein that was first isolated from *Clostridium pasteurianum* [39]. Fdxs coordinate FeS clusters by Cys ligands and can be divided into different groups based on the number and types of FeS clusters [40]. Fdxs that participate in electron transport to Fe protein are found to have [2Fe-2S]-type [41], [4Fe-4Fe]/[3Fe-4S]-type, or 2[4Fe-4S]-type clusters [42–44]. The [2Fe-2S]-type is normally found in cyanobacteria with the corresponding gene named *fdxH*, while the other two are found in diverse diazotrophic groups and are usually designated as *fdxN*. The possible redox states and redox potentials of Fdxs are modulated by their peptide structure (and ultimately, their coordination spheres) and occur in the ranges of −0.24 to −0.46 V, −0.05 to −0.42 V, and −0.28 to −0.68 V, for the couples $[2Fe-2S]^{2+/1+}$, $[3Fe-4S]^{1+/0}$, and $[4Fe-4S]^{2+/1+}$, respectively (vs. SHE) [45]. In comparison to Fld, Fdx is more sensitive to $O_2$ due to the lability of their FeS clusters. Phylogenetic analysis indicates that Fld (NifF) is enriched in diazotrophs with aerobic or facultative anaerobic life styles, and is believed to be an adaptive strategy for the diversification of Nif-nitrogenase from anaerobic to aerobic taxa during evolution [46]. Although other Fdxs are found to be involved in the assembly of nitrogenase cofactors or for protecting it against oxygen [47,48], the following discussion focuses on Fdxs involved in electron transport.

Fld forms a tight complex with Fe protein with high affinity (Figure 2). Reported dissociation constants for the pairs from *Klebsiella pneumoniae* are 13 μM (MgATP-bound Fe protein) and 49 μM (MgADP-bound Fe protein) and from *Rhodobacter capsulatus* is 0.44 μM. [49,50] It is generally believed that in the Fe protein cycle, reduction of Fe protein and exchange of MgADPs for MgATPs takes place after its dissociation from MoFe protein, implying that the oxidized Fe protein cannot be reduced when bound to MoFe protein. Further, the Fe protein is believed to interact with Fld/Fdx using the same binding interface that is used with the MoFe protein [16,33,51,52]. This hypothesis is guided by docking models that predict electrostatic interactions between the positively charged surroundings of the [4Fe-4S] cluster of Fe protein and the negatively charged surroundings of FMN of NifF from *A. vinelandii*, possibly assisted by an eight amino acid loop (residues 64–71) on NifF [16,32,52] (Figure 2). The MgADP-bound state of the Fe protein has the most complementary docking interface with Fld compared with the MgATP-bound state. Experimentally, this hypothesis is supported by results from cross-linking studies and time-resolved limited proteolysis using NifF and Fe protein from *A. vinelandii* [16,52]. Similar modeling results were obtained for Fdx, although experimental support has not yet been reported [33,41].

Early studies indicated that the specific catalytic activity of MoFe protein [27,51] and ATP/$e^-$ efficiency of nitrogenase are higher when NifF is used as the reductant of Fe protein rather than DT. Second-order rate constants of Fe protein reduction are two to three orders of magnitude times higher when NifF is used [53] or included [16] as reductant compared to DT [16] or Ti (III) [53]. Both physiological reductants and chemical reductants reduce MgADP-bound Fe protein faster than its nucleotide-free form. Nevertheless, recent reexamination of the rate difference under pseudo-first order reaction conditions showed similar trends [16]. The diminished performance of DT might be partially due to the slow generation of radical anion $SO_2^{\bullet-}$ ($K_d$ 1.5 nM, rate constant of ~2 $s^{-1}$) [16].

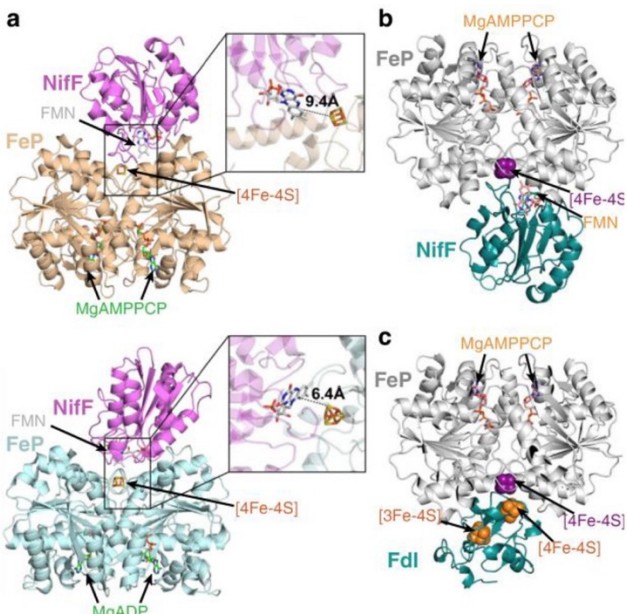

**Figure 2.** Docking models for reduction of the Fe protein by NifF and Fdx (FdI). (**a**) MgAMPPCP-Fe protein (top, PDB: 4WZB) and MgADP-Fe protein (bottom, PDB:1FP6) interacting with NifF (PDB:1YOB). (**b**) MgAMPPCP-Fe protein (PDB:4WZB) interacting with NifF (PDB:5K9B) and (**c**) FdI (PDB:6FDR). Hypothetical electron transfer distances between the [4Fe-4S] cluster of the Fe protein (NifH) and the FMN cofactor of NifF are shown. Reprinted (adapted) with permission from [9]. Copyright 2020 American Chemical Society. Republished with permission of American Soc for Biochemistry and Molecular Biology Copyright 2017, from [52]; permission conveyed through Copyright Clearance Center, Inc. Adapted with permission from [33]. Copyright 2017 John Wiley and Sons.

The findings above help revise kinetic models of the Fe protein cycle, as the Fe protein reduction rate has been used to estimate its dissociation from MoFe protein. When using DT, the complex dissociation rate was measured as $\sim 6$ s$^{-1}$ [11,51] and was believed to be the limiting step of the Fe protein cycle. Yet a high rate of 759 s$^{-1}$ was obtained when NifF was used, indicating that P$_i$ release (25–27 s$^{-1}$) is actually the rate-limiting step [16]. In line with this, the specific activity of nitrogenase was shown to increase by 50–170% with the presence of NifF as compared to DT alone [16,51].

Electron transfer efficiency to nitrogenase is expected to be improved when using physiological electron donors. While the $1e^-$ reduced [4Fe-4S]$^{1+}$ of Fe protein is commonly believed to be the only physiologically relevant state [6], Watt and Reddy discovered that the [4Fe-4S]$^{1+}$ cluster can be further reduced by $1e^-$ to a stable all-ferrous [4Fe-4S]$^0$ state by methyl viologen (MV) [54]. An $E^{o\prime}$ of $-0.46$ V vs. SHE for the 1+/0 couple was reported [54]. Since, many studies have characterized the [4Fe-4S]$^0$ state of Fe protein reduced by Ti(III) or Eu(II), Eu(II) complexes employed to study nitrogenase typically have $E^{o\prime}$'s ranging from $-0.6$ to $-1.1$ V vs. SHE [53,55–61]. One study reported an $E^{o\prime}$ of $-0.79$ V for the [4Fe-4S]$^{1+/0}$ couple [62]. Lowery et al. (2006) showed MgADP or MgATP-bound Fe protein can be reduced by FldHQ state of NifF (*A. vinelandii*) from [4Fe-4S]$^1$ to [4Fe-4S]$^0$ independent of catalysis, which supports a $E^{o\prime}$ of $-0.46$ V of the [4Fe-4S]$^0$ state and the possibility of its physiological relevance [63]. The Fe protein's [4Fe-4S]$^0$ state could allow two electrons to be transferred from FldHQ to Fe protein per two ATP molecules hydrolyzed, reaching a 1:1 ATP:e$^-$ ratio. This was indeed observed in a few studies using either NifF or Ti(III) [53,63,64]. Yet in most studies the ATP:e$^-$ ratio remained at 2:1, even when NifF was used as the reductant [9,16]. In contrast, DT only reduces the [4Fe-4S] cluster to the [4Fe-4S]$^{1+}$ state, and thus transfers only one e$^-$ per transient association cycle.

These findings call for further investigation into nitrogenase's natural electron donors. Identifying Fld/Fdx that directly transfers electrons to nitrogenase and to what extent could be difficult due to their redundancy in both genome and function; it is common for a bacterial or archaeal genome

to encode multiple Fld/Fdxs [65]. Due to the high energetic expense of $N_2$ fixation, nitrogenase genes, including the ones encoding for electron transport, are often co-located and/or transcriptionally co-regulated [66,67]. Mutagenesis combined with nitrogenase activity assays in cell extracts can provide direct proof of a gene's function in electron transfer to nitrogenase. However, most diazotrophs have more than two Fld/Fdxs capable of direct electron-transfer to nitrogenase (Table 1). In rare cases such as in *K. pneumoniae*, a sole Fld NifF is the electron donor and diazotrophic growth is abolished in *nifF* deletion mutants. In comparison, *A. vinelandii* is still able to grow diazotrophically (though much is undermined) when the two electron transport components *nifF* and *fdxA* are deleted [68]. In phototrophic bacteria, Fld is commonly found to serve as an electron donor under iron depleted conditions, whereas Fdx is the main electron donor under iron replete conditions [69].

**Table 1.** Electron transport components required for nitrogenase in representative diazotrophs. Gene names are shown with protein names, if available.

| Species | Direct Electron Donor to Fe Protein | Pathway for Reducing Electron Donor | Reference |
|---|---|---|---|
| *Anabaena* PCC 7120 | Fdx (*fdxB*, *fdxH*, *fdxN*) Fld (*nifF* [1]) | FNR (*petH*) Hydrogenase (*hupSL*) PFOR (*nifJ* [1]) | [70–79] |
| *Azotobater vinelandii* | Fdx (*fdxN*) Fld (*nifF*) | FixABCX (*fixABCX*) FNR Rnf1 (*rnf*) | [68,80–82] |
| *Clostridium pasteurianum* | Fdx | PFOR (*nifJ*) Hydrogenase | [83–85] |
| *Klebsiella pneumoniae* | Fld (*nifF*) | PFOR (*nifJ*) | [86–90] |
| *Sinorhizobium meliloti* | Fdx (*fdxN*) | FixABCX (*fixABCX*) | [91–93] |
| *Rhodobacter capsulatus* | Fdx (*fdxNC*) Fld (*nifF* [1]) | Rnf1 (*rnf*) Hydrogenase (*hupSL*) | [94–101] |
| *Rhodopseudomonas palustris* | Fdx (*fer1*, *ferN*) Fld (*fldA* [1]) | FixABCX (*fixABCX*) Hydrogenase (*hupSL*) | [69,102,103] |
| *Rhodospirillum rubrum* | Fdx (*fdxN*, *fdxI*) Fld (*nifF* [2]) | FixABCX (*fixABCX*) Hydrogenase (*hupSL*) PFOR (*nifJ* [2]) | [104–109] |

[1] These genes are expressed by cells growing under iron depleted conditions. [2] These genes are expressed by cells growing under anaerobic conditions in the dark.

## 2.2. Electron Transfer to Flavodoxin and Ferredoxin

The electrons transferred to Fld/Fdx directly come from pyruvate, NAD(P)H, or hydrogen. Five major enzyme systems capable of reducing Fld/Fdx have been identified (Figure 3, Table 1) and are discussed below.

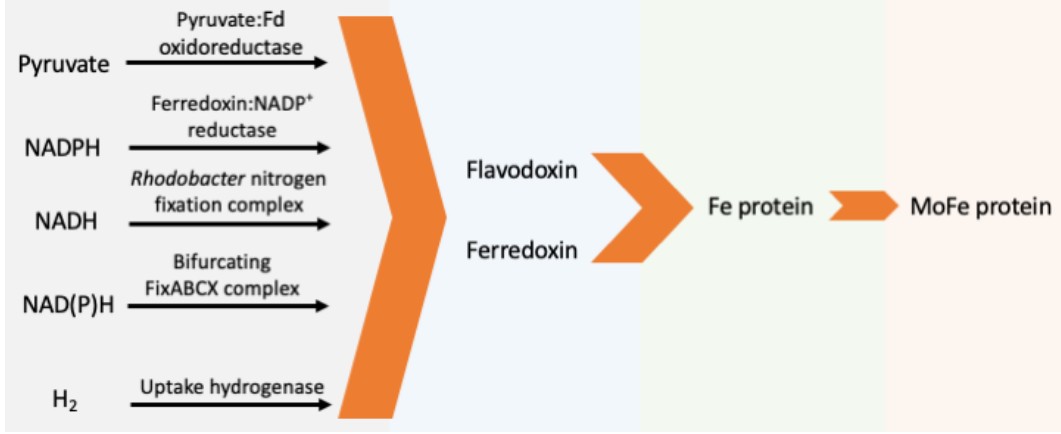

**Figure 3.** Schematic pathway of natural electron transfer to nitrogenase. (Fd: ferredoxin or flavodoxin).

Pyruvate-fld/fdx oxidoreductase (PFOR) catalyzes oxidization of pyruvate to acetyl-CoA and $CO_2$ with reduction of Fld or Fdx by using thiamine pyrophosphate and FeS clusters as cofactors. The $E^{o'}$ for the pyruvate cleavage is $-0.5$ V (vs. SHE). PFOR involved in electron transport to nitrogenase is commonly found as part of *nif* operon designated as *nifJ*. The deletion of *nifJ* in *K. pneumoniae* was found to abolish diazotrophic growth [86–88].

Fdx-NADP$^+$ Reductase (FNR) catalyzes the reversible reaction of Fdx oxidation with NADP$^+$ reduction by a flavin adenine dinucleotide (FAD) cofactor. Fdxs can possess $E^o$'s more negative than NADPH/NADP$^+$ ($-0.34$ V vs. SHE), and the forward reaction (*i.e.*, Fdx oxidation) is energetically favorable. Yet, FNRs that favor the reverse reaction are found in *A. vinelandii* and cyanobacteria and are thought to support nitrogenase activity based on enzyme assay and gene coregulation [70,71,80].

Rhodobacter Nitrogen Fixation (Rnf) complex is a transmembrane ferredoxin—NAD$^+$ oxidoreductase, which catalyzes the NADH-dependent reduction of Fld/Fdx with the a depletion of electrochemical gradient [110]. The Rnf complex shares homology with Na$^+$ pumps and is distributed among diverse N$_2$-fixing and non-nitrogen-fixing microbes [29,110]. The disruption of genes in the *rnf* operon of *R. capsulatus* resulted in a significant decrease in nitrogenase activity and abolished its ability to grow diazotrophically [94,95].

Bifurcating FixABCX couples the endergonic reduction of Fld to the exergonic reduction of coenzyme Q ($E^{o'} = +0.01$ V vs. SHE) (then to respiratory chain) at the cost of NADH by using a flavin at the bifurcating site [81,82,91,104]. This membrane protein complex was identified to have three FAD moieties, one each in FixA, FixB, and FixC, and two [4Fe-4S] clusters in FixX [81]. Disrupting the Fix system in *Rhodopseudomonas palustris*, *Rhodospirillum rubrum*, and *Sinorhizobium meliloti* completely abolishes or significantly impairs their ability to grow under N$_2$ fixing conditions [92,102,104]. A double mutant of Fix and RNF1 abolished diazotrophic growth in *A. vinelandii* [81].

Uptake hydrogenase (hup) is a heterodimer of the *hupA* and *hupB* (or *hupS* and *hupL*) gene products that has either a Ni-Fe or Fe-Fe active center [111]. It recaptures H$_2$ produced during N$_2$ fixation (Equation (1)) for use as electron donors cycled back to nitrogenase [96], and also seems to protect nitrogenase from oxygen, possibly by catalyzing oxyhydrogen reaction [112,113]. Overproduction of H$_2$ from nitrogenase was achieved by deleting the uptake hydrogenase [72,114]. The *hup* genes are co-regulated with nitrogenase through complicated and diverse regulatory pathways [83,84,97,103], which was exploited to identify H$_2$-overproducing variants of the Fe protein [115].

The diverse strategies to reduce Fld/Fdx by different microbes can be better understood if put into metabolic context (Table 1). Bioinformatic analysis indicates proteins that reduce Fld/Fdx with H$_2$ or pyruvate are enriched in anaerobes, while those with NADH/NADPH are enriched in aerobes, facultative anaerobes, and anoxygenic phototrophs [116]. Aerobic, facultatively anaerobic, and anoxygenic phototrophic diazotrophs produce NADH/NADPH, which is not considered reducing enough to provide electrons for nitrogenase; thus, FixABCX and Rnf are acquired to generate low potential electrons [46]. Specifically, oxygenic phototrophs can energize electrons to low potential using photosystem I (PSI), but the low potential electrons are not available for nitrogenase, as the latter must be spatially (forming heterocysts) or temporally (under the control of circadian clocks) separated from PSI, in which case FNR is used [70]. Interestingly, some non-heterocystous cyanobacteria show nitrogenase activity only under light conditions, implying the existence of molecular mechanisms to protect nitrogenase against oxygen evolved by photosystems [113,117,118].

## 3. Electron Transfer to Nitrogenase in Engineered Biological Systems

Developing and improving biological systems for N$_2$ fixation is greatly motivated by the need to design sustainable and economic solutions to nitrogen limitation of crop productivity. Decentralization of NH$_3$ production is considered one possible strategy to improve environmental sustainability. Great efforts have been made towards engineering plant-associated microbes or plants themselves to heterologously express nitrogenases [4]. From the perspective of synthetic biology, genes required for functional nitrogenase can be grouped into modules of structural genes, genes involved in biosynthesis

and maturation of cofactors, and electron-transport components (ETC) [119]. ETC include Fld/Fdx and the reductase of Fld/Fdx. ETC can come from the native host to nitrogenase or be substituted by homologues from the expression host, if compatible (Table 2). In general, the former results in higher nitrogenase activity, while the latter gives the advantage of expressing fewer genes and simplifying genetic engineering [120,121]. It is important to bear in mind that there is limited crosstalk between Fld/Fdx and the Fe protein, and between Fld/Fdx and their reductases from different hosts. For example, Fdx from either *Clostridium pasteurianum* or *R. rubrum* were ineffective in coupling pyruvate oxidation to nitrogenase activity in the cell lysate of *K. pneumoniae* NifF mutant. NifF from *A. vinelandii* was only one-third as effective as NifF from *K. pneumoniae* at transferring electrons from PFOR to *Klebsiella* nitrogenases [88]. Further, in vivo activity could not be predicted by in vitro assays. For example, Fdxs from *R. capsulatus* and *S. meliloti* equally support in vitro acetylene ($C_2H_2$) reduction to ethylene ($C_2H_4$) by *R. capsulatus* nitrogenase, yet heterologous in vivo complementation by each other's Fdx was unsuccessful [122,123].

**Table 2.** Summary of heterologous expression of active nitrogenase in prokaryotic hosts.

| Expression Host | Source of Nitrogenase Structural and Accessory Genes [1] | Genes Encoding for Electron Transport Component (ETC) to Nitrogenase | Source of ETC | Reference |
|---|---|---|---|---|
| *E. coli* JM109 | *A. vinelandii* DJ [2] | *nifFJ* *fldA, ydbK* | *K. oxytoca* M5al *E. coli* JM109 | [124] |
| *E. coli* JM109 | *K. oxytoca* M5a1 | *nifFJ* | *K. oxytoca* M5a1 | [125] |
| *E. coli* JM109 | *Paenibacillus* sp. WLY78 | *fldA, ydbK* | *E. coli* JM109 | [121] |
| *E. coli* JM109 | *Paenibacillus* sp. WLY78 | *nifFJ* *fer* or *fldA, pfoAB* | *K. oxytoca* M5al *Paenibacillus* sp. WLY78 | [120] |
| *E. coli* MG1655 | *K. oxytoca* M5al | *nifFJ* | *K. oxytoca* M5al | [126] |
| *Pseudomonas protegens* Pf-5 | *Pseudomonas stutzeri* A1501 | *nifFJ* | *Pseudomonas protegens* Pf-5 | [127] [3] |
| *Pseudomonas protegens* Pf-5 | *A. vinelandii* DJ | *nifF, fixABCX, rnf1* | *A. vinelandii* DJ | [128] [3] |
| *Pseudomonas protegens* Pf-5 | *P. stutzeri* A1501 | *nifF, fdxN, rnf* | *Pseudomonas protegens* Pf-5 | [128] |
| *Rhizobium* sp. IRBG74 | *K. oxytoca* M5al | *nifFJ* | *K. oxytoca* M5a1 | [128] |
| *Rhizobium* sp. IRBG74 | *R. sphaeroides* 2.4.1 | *rnf* | *Rhizobium* sp. IRBG74 | [128] |
| *Synechocystis* sp. PCC 6803 | *Leptolyngbya boryana* strain dg5 | *fdxH* *petFH* | *Leptolyngbya boryana* strain dg5 *Synechocystis* sp. PCC 6803 | [118] |
| *Synechocystis* sp. PCC 6803 | *Cyanothece* sp. ATCC 51142 | *fdxNHB, petFH* | *Synechocystis* sp. PCC 6803 | [113] |

[1] MoFe-type nitrogenase (*nif*) was expressed in these studies unless otherwise noted. [2] Structural genes of Fe-only nitrogenase were expressed in combination with accessory genes of MoFe-type nitrogenase. [3] Other combinations of *nif* genes and expression hosts were explored in this study but are not listed here.

By now, there has been greater success in expressing active nitrogenases in prokaryotic hosts (Table 2). Since Dixon and Postgate conjugated $N_2$ fixation genes from *Klebsiella oxytoca* into *Escherichia coli* in 1972 [129], nitrogenases from various sources have been cloned and expressed in different hosts (Table 2). *E. coli* has been used as a platform to identify minimal gene requirements for active nitrogenase [121,124] and to optimize expression [125,126,130]. It was found that *fldA* (homologue to *nifF*) and *ydbK* (homologue to *nifJ*) in *E. coli* can support nitrogenase activity, although activity is significantly improved when *nifFJ* from diazotrophs are employed [121,124]. Non-diazotrophic,

oxygenic cyanobacteria have also been explored as expression hosts, as they serve as a simpler model of plant chloroplasts [113,118]. The $O_2$ tolerance of nitrogenase was enhanced by co-expressing an uptake hydrogenase in *Synechocystis* sp. PCC 6803 [113]. Epiphytic and endophytic bacteria are desirable platforms for $N_2$ fixation, as they can be directly applied. Challenges here remain in the fact that different plants have specific colonizers, and that most natural diazotrophic endophytes do not express nitrogenase under the desired conditions or to a desired level [131,132]. A recent study by Ryu et al. applied different strategies to a wide range of hosts and engineered inducible promoters in combination with nitrogenase genes to tackle both of those problems [128]. Overall, heterologous nitrogenases showed significantly lower activities compared to that in their native hosts with the exception of MoFe nitrogenase from *K. oxytoca* expressed in *E. coli* [125,126,128].

Expressing nitrogenases in eukaryotic systems—either plant or yeast and green algae—as simpler models, have limited success [4]. Separate Nif components have been expressed in these systems but the formation of a fully functional nitrogenase has not been achieved [4]. Both mitochondria and chloroplasts are candidate organelles to host the expression of nitrogenase. The former provides MgATP and a low $O_2$ environment due to active respiration, while the latter produces abundant reduced Fdx and MgATP by photosynthesis with the byproduct of $O_2$ [4,133]. Several lines of evidence suggest that chloroplasts might be a more suitable choice. While both locations have Fdx-FNR-type electron transport modules, the ones in chloroplasts function as part of the photosynthesis pathway and share more similarity to those in diazotrophic bacteria. In mitochondria, the Fdx-FNR homologue modules are called adrenodoxin and NADPH-dependent adrenodoxin oxidoreductase as part of biosynthesis pathway of biotin [134–136]. Yang et al. (2017) used *E. coli* as a chassis to study the compatibility between MoFe and FeFe nitrogenases with ETC modules from plant chloroplasts, root plastids, and mitochondria. They found that Fdx-FNR from chloroplasts and root plastids can support the activities of both types of nitrogenase, while the ETC module from mitochondria could not. A hybrid module of mitochondrial FNR and the cyanobacteria Fdx could support nitrogenase activities [137]. In addition, chloroplast genomes of some plants and algae encode a nitrogenase-like enzyme called dark-operative protochlorophyllide oxidoreductase (DPOR) that participates in biosynthesis of chlorophylls and is also $O_2$ sensitive [138]. It was demonstrated that the Fe protein from *K. pneumoniae* nitrogenase can be expressed and functionally substitute for its homologue in DPOR in *Chlamydomonas reinhardtii,* suggesting that chloroplasts have the potential to provide a suitable environment for nitrogenase [139,140].

Alternative strategies have been explored to increase nitrogenase activities using natural diazotrophs. Several studies exploited a MoFe nitrogenase variant ($\alpha$-V70A, $\alpha$-H195Q) [141] to catalyze the reduction of $CO_2$ to methane as a way for in vivo production of biofuels in *R. palustris*. While ATP is supplied by cyclic photophosphorylation, it was found that the electron flow to nitrogenase can be enhanced by manipulating metabolic pathway or state, such as providing cells with organic alcohols, diverting electrons away from biomass synthesis by using nongrowing cells, or blocking the Calvin–Benson–Bassham cycle [142,143]. Further, the same group demonstrated that wild-type Fe-only nitrogenase in *R. palustris* reduces $CO_2$ simultaneously with nitrogen fixation to yield $CH_4$, $NH_3$, and $H_2$. Excitingly, this seems to be a universal feature for the Fe-only nitrogenases. The amount of $CH_4$ produced was low but sufficient to support the growth of an obligate methanotroph in co-culture with oxygen added at intervals [143]. Further studies are needed to see whether the above strategies of metabolic engineering can be applied to enhance the activity of Fe-only nitrogenase and whether similar principles can be applied to engineer other diazotrophs.

Liu et al. demonstrated the use of $H_2$ generated from catalytic water splitting driven by renewable energy to support diazotrophic growth of autotroph *Xanthobacter autotrophicus*. In that way the production of *X. autotrophicus* as biofertilizer or $NH_3$ (when glutamine synthetase was inhibited) is efficiently connected to atmospheric nitrogen. The biomass produced was applied to radishes and significantly increased the yield of radish storage roots (Figure 4) [144]. Their system achieved a

nitrogen reduction turnover numbers of $\sim 9 \times 10^9$ bacterial cell$^{-1}$ and $2 \times 10^6$ nitrogenase$^{-1}$ and a turnover frequency of $1.9 \times 10^4$ s$^{-1}$ per bacterial cell, or $\sim 4$ s$^{-1}$·nitrogenase$^{-1}$.

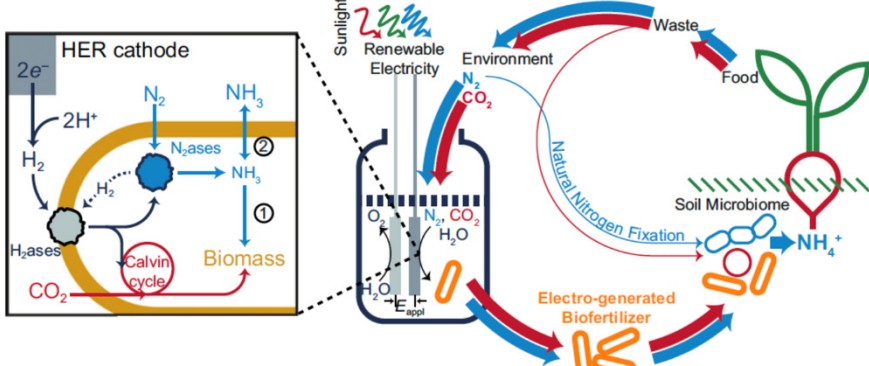

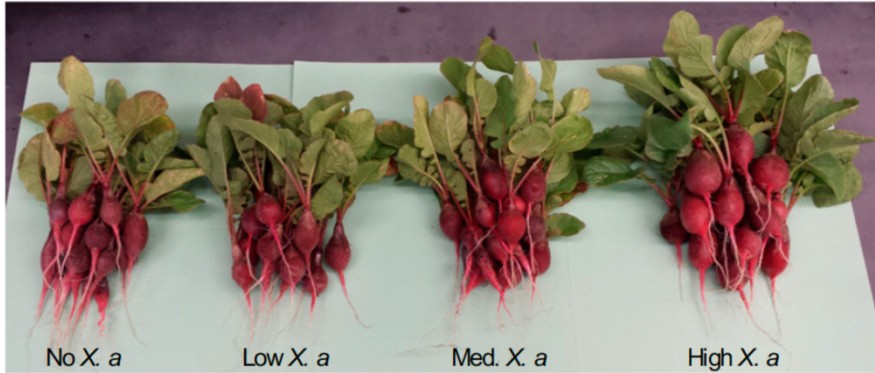

**Figure 4.** (**Top**) Schematic of ammonia (NH$_3$) production in a biohybrid system that produces hydrogen (H$_2$) from renewable electrical energy and sunlight. Hydrogenases within *Xanthobacter autotrophicus* subsequently oxidize the H$_2$ to ultimately supply electrons to nitrogenase for dinitrogen (N$_2$) fixation. The generated NH$_3$ is incorporated in biomass (pathway 1) or can diffuse extracellularly by inhibiting biomass formation (pathway 2). Red arrows represent carbon cycling and blue arrows represent nitrogen cycling; line widths represent the relative fluxes of these pathways. (**Bottom**) Enhanced radish growth upon biofertilization with *X. autotrophicus* grown from electricity/sunlight-sourced H$_2$. Reproduced from [144] with permission.

## 4. In Vitro Electron Transfer to Nitrogenase

### 4.1. Fe protein-Dependent Activity

For in vitro nitrogenase activity, research has sought to artificially deliver electrons to the Fe protein. Subsequently, the reduced Fe protein transfer electrons to the MoFe protein in the presence of the MgATP that is required for the Fe protein cycle. As discussed above, the most commonly employed electron donor is DT, which supports the formation of the 1e$^-$-reduced [4Fe-4S]$^{1+}$ state of the Fe protein. While DT supports nitrogenase catalysis, it can be undesirable in many ways: (i) it is considered single-use since its regeneration is complex, (ii) DT is not particularly stable in aqueous solutions, and (iii) the reduction potential of DT is dependent on its concentration and pH, which can complicate thermodynamic and kinetic interpretations of metalloenzyme properties [30].

In addition to DT, other electron mediators employed with the Fe protein include MV (and derivatives), Ti(III) citrate, Eu(II) complexes, and one of the presumed in vivo electron donors, Fld (NifF) [9]. While Ti(III) and Eu(II) complexes are useful due to their low reduction potentials, they are typically prepared as a stock of a single-use reductant. In the case of Ti(III), Seefeldt and Ensign reported in 1994 that Fe protein reduction could occur such that catalysis by the MoFe protein could be supported, where similar rates of H$^+$ and C$_2$H$_2$ reduction were observed in comparison to DT-driven

assays [145]. Further, the oxidation of Ti(III) to Ti(IV) could be followed spectrophotometrically. MV and NifF have also been employed as electron donors to the Fe protein (for subsequent catalysis); however, they are typically employed alongside a relatively higher concentration of DT as the reducing agent [9,16]. As mentioned above, Yang et al. demonstrated that the reduction of nucleotide-free Fe protein by DT increased around 10-fold when NifF was included, presumably as mediator for DT reduction of the Fe protein [16]. Additionally, it was observed that the presence of either MgADP or MgATP diminished DT-driven Fe protein reduction by ~100-fold although this was alleviated by the inclusion of NifF. $H_2$, NADH, and $KBH_4$ have also been employed to transfer electrons to MV prior to catalysis by nitrogenase (in cell-free extracts of *C. pasteurianum*), although in all of the reported cases the oxidation of MV by nitrogenase cannot be followed, since the oxidized MV is regenerated in these assays [146,147].

Electrochemical Methods

Recently, MV has been utilized as the sole reductant for Fe protein-dependent nitrogenase activity assays. In 2017, Milton et al. demonstrated the use of electrochemically reduced MV (in the $1e^-$-reduced $MV^{\bullet+}$ state) as the sole electron donor to the Fe protein [148]. First, it was demonstrated that the oxidation of MV could be followed spectrophotometrically. This was employed to rapidly determine the optimal Fe:MoFe protein ratio for MV-dependent assays (~20:1), although it was also observed that increasing MV concentrations resulted in diminished nitrogenase activity [146,148]. The nature of this has been recently ascribed to the dimerization of MV [149]. In addition to spectrophotometric activity, an electrochemical method was also employed to reduce MV in situ for nitrogenase activity assays [148]. First, this paves the way for the utilization of renewable electrical energy for "bioelectrosynthetic" $N_2$ reduction under ambient temperature and pressure (Figure 5).

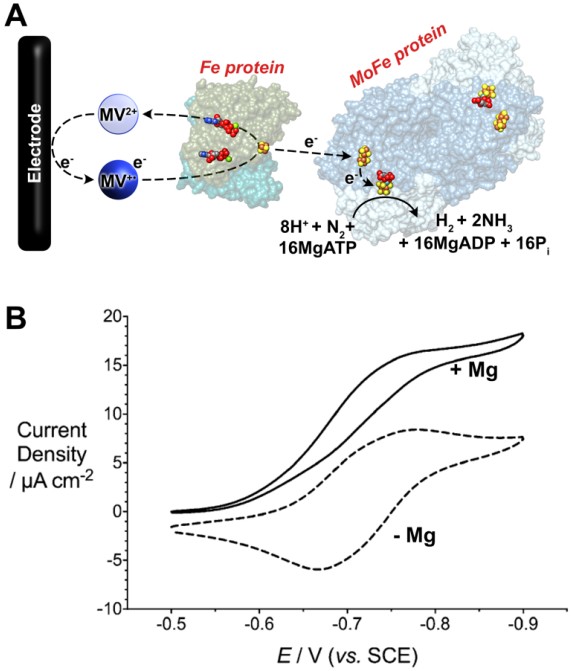

**Figure 5.** (**A**) Bioelectrocatalytic dinitrogen ($N_2$) fixation by Mo-dependent nitrogenase using methyl viologen (MV) as an electron mediator. (**B**) Cyclic voltammetric data (scan rate = 0.002 V $s^{-1}$) for bioelectrocatalytic $N_2$ fixation. The electrochemical cell contained MOPS buffer (100 mM, pH 7), a MgATP-regenerating mixture (ATP, creatine phosphate, creatine phosphokinase), 0.1 mg $mL^{-1}$ of MoFe protein, and 20× equivalents of the Fe protein. The addition of $Mg^{2+}$ permits MgATP hydrolysis by the Fe protein and initiates bioelectrocatalytic $N_2$ fixation. Reprinted (adapted) with permission from [150]. Copyright 2019 American Chemical Society.

This method was capable of electrochemically driving $N_2$ reduction to $NH_3$ where 59% of the electrons delivered to nitrogenase were directed towards $N_2$ fixation. This initial example reported $NH_3$ production at a rate of 35 nmol $NH_3$ min$^{-1}$ mg$^{-1}$ MoFe protein, corresponding to a turnover frequency (TOF) of 4.2 (single time point). In this section, TOF is calculated to be the number of moles of product formed per mole of MoFe protein (assuming, for consistency, a mass of 240 kDa for the MoFe protein) per minute. Second, the current recorded at the electrode corresponds to the rate and magnitude of electron consumption by nitrogenase (and thus, substrate reduction), which provides a method to study substrate reduction by nitrogenase in real time [151]. The subsequent optimization of this MV electrochemical approach resulted in an observed rate constant for electron flux through nitrogenase when fixing $N_2$ approached the expected value of 13 s$^{-1}$ (reported as 14 s$^{-1}$), where rate-limiting electron transfer at 13 s$^{-1}$ corresponds to an optimal TOF of 195 (accounting for *re* of $H_2$). Further, MV (and derivates) can also serve as an efficient electron donor to other (metallo)enzymes, such as formate dehydrogenases and hydrogenases, either for $H_2$/formate ($HCOO^-$) oxidation or for $H^+$/$CO_2$ reduction [152]. Of particular interest is the ability of MV to mediate electrons or enzymatic NADH formation [153,154]. To this end, the Minteer group has extended the use of reduced MV in bioelectrosynthetic cells to reduce NADH. In this way, the $NH_3$ formed by nitrogenase can be upgraded to further products of interest, such as chiral amines and chiral amino acids [155,156].

### 4.2. Fe Protein-Independent Activity

While nitrogenase fixes $N_2$ under considerably milder conditions than the Haber–Bosch process (i.e., at ambient pressure and temperature), nitrogenase can still be considered to be energy-intensive due to hydrolysis of at least 16MgATP for the fixation of each equivalent $N_2$. The hydrolysis of MgATP corresponds to around −50 kJ mol$^{-1}$ in vivo; thus, there is significant interest in decoupling the Fe and MoFe proteins [20]. Not only does this present the possibility of MgATP-independent $N_2$ fixation, but catalytic rates could also be improved, given that the rate-limiting step of $N_2$ fixation is associated with the Fe protein [16]. As outlined above, evidence to supporting the idea that electron transfer between the Fe and MoFe proteins occurs prior to MgATP hydrolysis suggests that artificial reduction of the MoFe protein independent of the Fe protein (and therefore MgATP independent) could be possible. Since 2010 (when two papers submitted within two weeks of each other reported Fe protein-independent MoFe protein catalysis), chemical, photochemical, and electrochemical techniques have sought to deliver electrons to the MoFe protein for substrate reduction; prominent examples are covered below.

### 4.2.1. Chemical Methods for Electron Transfer

In 2010, Danyal et al. reported substrate reduction by the MoFe protein from *A. vinelandii*, for which low-potential Eu(II) was employed as the artificial reductant [157]. Eu(II) was prepared by bulk electrolytic reduction of a $Eu_2O_3$ ($E^{o'}$ = −0.36 V vs. SHE) solution, followed by the addition of a chelating polyaminocarboxylate ligand that lowered the potential of the pre-reduced Eu(II) center −0.88 or −1.14 V vs. SHE (depending on the ligand employed, reported at pH 8) [60,157]. Thus, electrochemistry was employed to prepare the reductant in batch mode, and this example can be treated as a "chemical" reduction approach. While the authors noted that $N_2$ fixation was not observed, the 2e$^-$-reduction of hydrazine ($N_2H_4$) to $NH_3$ was observed; further, a ß-Y98H MoFe protein mutant was required for the formation of significant quantities of $NH_3$ when compared to blank/control experiments. The ß-Y98 residue, located between the P cluster and FeMo-co, was previously identified as a residue that is important to electron transfer [158]. A TOF of 41 was reported for the system; further, $N_2H_4$ is an important substrate where its 2e$^-$-reduction to $NH_3$ may represent an intermediate step of $N_2$ fixation by nitrogenase. In 2015 Eu(II) complexes were coupled with two additional MoFe mutants, ß-F99H and $\alpha$-Y64H, which also revealed enhanced $N_2H_4$ reduction in comparison to the WT MoFe protein [159]. The $\alpha$-Y64H mutation was found to improve to TOF to 72. It was also demonstrated that these MoFe protein mutants could reduce azide ($N_3^-$) to $NH_3$, albeit it with a reduced TOF of 2 (single time point). However, only the originally reported ß-Y98H MoFe protein was found to reduce

H⁺ to H₂ when coupled with a Eu(II) ethylenediaminetetraacetic acid (EDTA) complex. Of the Eu(II) complexes tested within this study, the Eu(II)–EDTA complex has the mildest reduction potential (−0.84 V vs. SHE). However, the Eu(II)–EDTA complex was selected, as minimal H₂ is evolved when using this Eu(II) complex in the absence of MoFe proteins. Thus, one explanation for the inability of the ß-F99H and α-Y64H proteins to reduce H⁺ could be that the $E^{o'}$ of Eu(II)–EDTA complex was not reducing enough. Further, H₂ evolution was measured to be ~7 nmol H₂ min⁻¹ mg⁻¹ MoFe (ß-Y98H) which corresponds to a TOF of <2 (single time point). In contrast, Fe protein-derived electron transfer with a rate constant of ~13 s⁻¹ loosely corresponds to the formation of ~1625 nmol H₂ min⁻¹ mg⁻¹ MoFe and a TOF of 390 (H₂).

 In 2012, Lee et al. further demonstrated the ability of Eu(II) complexes to catalyze substrate reduction by nitrogenase [160]. However, substrate reduction was performed by variant MoFe proteins that lacked the FeMo-co and/or contained P cluster precursors (2[4Fe-4S] clusters). In *A. vinelandii*, the deletion of *nifB* leads to a MoFe protein that lacks the FeMo-co; instead, the deletion of *nifH* (Fe protein) leads to a MoFe protein that lacks the FeMo-co and has immature P-clusters (P*, 2[4Fe-4S] pairs) [6]. Remarkably, it was demonstrated that MoFe protein lacking FeMo-co could catalyze the reduction of H⁺, C₂H₂, C₂H₄, N₂H₄, cyanide (CN⁻), carbon monoxide (CO), and CO₂ to products including, H₂, NH₃, and alkanes/alkenes spanning C₁–C₇ [160]. TOFs for this system reached maxima of 0.02 (for CH₄ production from CN⁻, single time point) and 0.17 (for NH₃ production from CN⁻, single time point). Interestingly, the authors observed a ~2.5-fold increase in activity when using the Δ*nifH* MoFe protein vs. the Δ*nifB* MoFe protein; Jimenez-Vicente et al. recently demonstrated that the Fe protein of the V-nitrogenase system (encoded by *vnfH* and typically repressed in the presence of Mo) can substitute the NifH Fe protein in Δ*nifH* strains, which could explain the observed elevated activities [161]. In addition to the Mo-nitrogenase system, Eu(II) complexes have also been utilized to deliver electrons to the alternative V-nitrogenase system for MgATP-independent substrate reduction [162]. In 2015, Rebelein et al. demonstrated that Eu(II)-DTPA ($E^{o'}$ = −1.1 V vs. SHE, DTPA = diethylenetriaminepentaacetate) could function as an electron mediator to the VFe protein for CO₂ reduction [163]. In addition to the formation of CO and CH₄ formation (with a TOF for CH₄ production reaching $0.4 \times 10^{-3}$), C₂–C₄ hydrocarbons were also produced.

## 4.2.2. Photochemical Methods for Electron Transfer

 In 2010, another approach for MoFe protein reduction was reported by the Tezcan group, which employed a Ru-based photosensitizer to deliver electrons to the FeMo-co through the P cluster [164]. [Ru(bpy)₂(phen)]²⁺ possesses a long-lived photoexcited-state (*Ru^II), which upon quenching, results in the generation of a reducing Ru^I species that is not expected to be too dissimilar to the related [Ru(bpy)₃]⁺ complex with an $E^{o'}$ of around −1.28 V vs. SHE [9,164,165]. The authors prepared a Cys-reactive Ru complex which was subsequently attached to a Cys mutation introduced in proximity to the P cluster (α-L158C). In this way, the Ru photosensitizer could be placed ~15 Å away from the P cluster (important for efficient electron transfer rates)—similar to the location of the Fe protein's [4Fe-4S] cluster during MoFe protein association [9]. Wild-type MoFe protein has a single solvent-exposed Cys residue; it was hypothesized that the Ru-attachment to this residue would not yield significant photo-excited electron transfer and MoFe protein catalysis. Indeed, significant substrate-reduction activity was observed following the introduction of the α-L158C mutation, where the authors first observed H⁺ and C₂H₂ reduction at approximately 14 nmol H₂ min⁻¹ mg⁻¹ MoFe protein and 16 nmol C₂H₄ min⁻¹ mg⁻¹ MoFe protein (both being 2e⁻-reductions), corresponding to TOFs of ~3.4 and ~3.8 respectively. In 2012, the authors expanded on this approach and demonstrated that photosensitized MoFe protein can also facilitate the 6e⁻-reduction of CN⁻ to CH₄ with a TOF of <0.1 [166]. However, this approach was not able to produce NH₃ from N₂ fixation at detectable quantities.

 In 2016, a second photochemical approach for MoFe protein reduction was reported in which CdS nanorods were mixed with wild-type MoFe protein and N₂ fixation to NH₃ was observed [167]. In contrast to the low reduction potential afforded by the Ru-photosensitizer system, CdS nanorods

offer a milder $E^{o'}$ of $-0.8$ V vs. SHE, which can also be accessed upon illumination with visible light. Under $N_2$, this system was found to produce $NH_3$ at 315 nmol min$^{-1}$ mg$^{-1}$ MoFe protein which corresponds to ~63% of $NH_3$ production by the Fe protein-driven, ATP-dependent activity of the MoFe protein and a TOF of 75. Recently, Harris et al. expanded on this approach by investigating electrostatic interactions between the MoFe protein and the surface of CdS nanorods modified by charged functional groups (Figure 6) [168]. In addition, the authors also prepared a flexible linker to enable the attachment of the MoFe to the CdS nanorods by a Cys residue introduced near to the P cluster ($\alpha$-L158C). For the optimal configurations reported, maximal $H_2$ evolution rates of approximately 1250 nmol $H_2$ min$^{-1}$ mg$^{-1}$ MoFe protein were observed, which correspond to TOFs of approximately 300. In this example, $N_2$ fixation to $NH_3$ by the CdS nanorod biohybrid was not reported. In summary, precisely how the CdS biohybrid system bypasses the Fe protein and affords close-to-native TOFs is unknown. Yet, the ability to achieve $N_2$ fixation is promising and highly attractive to future ATP-independent $NH_3$ biotechnologies.

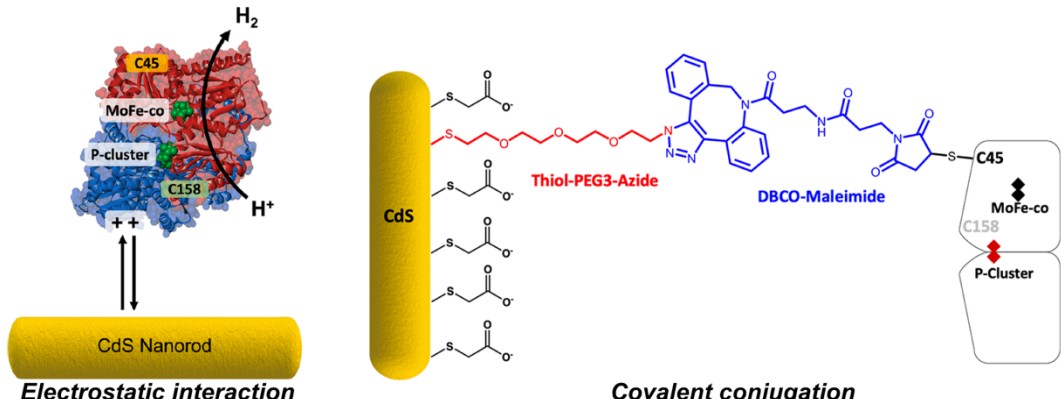

**Figure 6.** Harris et al. investigated the attachment of the MoFe protein to CdS nanorod surfaces: (**left**) surface-capping groups with different charges were employed, and (**right**) solvent-exposed Cys residues of the MoFe protein were used to covalently conjugate the MoFe protein to CdS nanorod surfaces. Reprinted (adapted) with permission from [168]. Copyright 2020 American Chemical Society.

### 4.2.3. Electrochemical Methods for Electron Transfer

Since 2016, electrochemical approaches have gained significant interest for artificial MoFe protein reduction and catalysis. Milton et al. reported the immobilization of the MoFe protein within a polymer at an electrode surface [169]. In this approach, bis(cyclopentadienyl)cobalt(II) (cobaltocene$^{1+/0}$, $E^{o'} = -0.96$ V vs. SHE) was used as a low potential electron mediator to deliver electrode-derived electrons to the MoFe protein. In addition to $H^+$ reduction, $N_3^-$ and nitrite ($NO_2^-$) reductions were also observed. As observed by Danyal et al., the use of the ß-Y98H MoFe protein mutant resulted in enhanced catalytic currents (consistent with enhanced catalysis): Faradaic efficiencies (FEs) of 35% and 101% and TOFs of 12 and 40 were reported for $N_3^-$ and $NO_2^-$ reduction (single time points). This method provided a method by which electron transfer to (and thus, catalysis of) the MoFe protein could be observed in real time. Hu et al. also adopted this approach to demonstrate that the MoFe and FeFe proteins could reduce $CO_2$ to $HCOO^-$ with respective FEs of 9% and 32%. Khadka et al. also demonstrated that this approach could be employed to investigate the rate-limiting step of the MoFe when catalyzing $H^+$ reduction [170]. In this study, the cobaltocene concentration was titrated such that electron transfer to the MoFe protein was not limiting, and bioelectrocatalytic $H^+$ reduction to $H_2$ was evaluated as a function of catalytic current. Following the addition of increasing $D_2O$ fractions to the electrochemical setup, a decrease in the catalytic current was observed, and this was attributed to a kinetic isotope effect. This was further exemplified by the use of a range of MoFe protein mutants and FeMo-co-transporting/storage proteins. Supporting density functional theory (DFT) calculations led to the conclusion that the protonation of a M-H at the FeMo-co is the rate-limiting

step of $H_2$ evolution (within the MoFe protein component). In 2018, Cai et al. also employed this approach to investigate catalysis of the VFe protein for $H^+$ and $CO_2$ reduction [171]. In addition to unsubstituted cobaltocene, mono-cobaltocene ([Co(Cp)(CpCOOH)], $E^{o'}$ = −0.79 V vs. SHE) and di-carboxy cobaltocene ([Co(CpCOOH)$_2$], $E^{o'}$ = −0.65 V vs. SHE) derivatives were also investigated as electron mediators for the VFe protein. After the passage of 4 C of charge within an electrochemical cell that contained the VFe and protein and dicarboxy-cobaltocene, ~850 µmol $H_2$ was detected µmol$^{-1}$ VFe protein (corrected to control experiments). Upon the addition of bicarbonate ($HCO_3^-$) as the $CO_2$ source, the passage of 4 C of charge led to the formation of ~20 nmol $C_2H_4$ µmol$^{-1}$ VFe and ~30 nmol propene ($C_3H_6$) µmol$^{-1}$ VFe protein (corrected to control experiments).

An additional approach to electrochemical MoFe protein reduction was reported in 2016, wherein Paengnakorn et al. reported the reduction $H^+$ by the MoFe protein also entrapped within a polymeric matrix [61]. A mixture of three Eu complexes was employed to mediate electron transfer to the MoFe protein, with $E^o$'s spanning −0.63 to −1.09 V vs. SHE (at pH 8). Excitingly, this approach was coupled with an attenuated total reflectance infrared (ATR-IR) spectroelectrochemical cell such that substrate binding at the FeMo-co could be interrogated as a function of potential. Interestingly, the binding of CO to the FeMo-co (which does not typically affect $H^+$ reduction) was found to commence at potentials of ~<−0.7 V vs. SHE. $H_2$ formation was also confirmed by gas chromatography resulting in a TOF of 14 for the ß-F99H mutant MoFe protein; this mutant and the ß-Y98H mutant (TOF = 13) MoFe proteins showed elevated electrocatalytic currents over the wild-type MoFe protein (TOF = 7) (single time points).

In contrast to the use of electron mediators to deliver electrons to the MoFe protein, Hickey et al. reported on the ability to directly deliver electrons to the MoFe protein when immobilized at electrode surfaces [172,173]. This approach represents an exciting new approach by which the MoFe protein (and the VFe and FeFe proteins) can be explored in greater detail, as this permits the direct observation of the redox state and reduction potentials of enzymatic cofactors. This approach permitted $NH_3$ production with TOFs of around 0.1–0.6, alongside a TOF of 2.6 for $H_2$ formation (under Ar, single time point) [172]. Interestingly, this article also identified that the background production of $H_2$ by the electrode (at low potentials) is inhibitory to the fixation of $N_2$ by nitrogenase, which should be avoided in future electrochemical investigation of nitrogenase.

## 5. Conclusions

As discussed, nitrogenase is the only enzyme known to reduce $N_2$ to $NH_3$. Thus, many streams of research are underway to understand how nitrogenase fixes $N_2$ and to engineer (semi)biological systems for $NH_3$ production. The understanding of nitrogenase's $N_2$ fixation mechanism could lead to the design of new bio-inspired catalysts with improved efficiencies, or lead to biotechnologies that exploit nitrogenase's reactivity to produce $NH_3$ (such as directing renewable electricity towards MgATP-independent $N_2$ fixation by the MoFe protein). The ability of engineered plants or symbiotic systems to produce their own $NH_3$ also represents a significant milestone. As presented here, many pathways (both natural and engineered) exist, or are being developed, to deliver electrons to nitrogenase for $N_2$ fixation. Nevertheless, ambiguity surrounds the possibility of the Fe protein undergoing the transfer of $2e^-$ during each transient association with the MoFe protein, which has the potential to halve nitrogenase's MgATP hydrolysis requirement. Further, a range of natural and unnatural electron donors are being explored for electron delivery. In the case of natural electron donors, it is still necessary to improve the efficiencies of electron transfer systems in non-diazotrophic hosts to support higher nitrogenase activity. In artificial in vitro systems, it is unclear as to why certain systems support $H^+$ reduction by nitrogenase (representing necessary M-H formation at the FeMo-co) but $N_2$ is not observed. For example, how are photochemical methods (such as CdS nanorods) able to support $N_2$ fixation, while other photochemical methods (i.e., Ru-based photosensitizers) are not? Similarly, it remains unclear as to how mediated electron delivery to the MoFe protein (or VFe/FeFe proteins) cannot currently support $N_2$ fixation (although $H^+$, $CO_2$, $N_3^-$, and $NO_2^-$ reduction has been achieved), whereas direct electron transfer appears to facilitate $NH_3$ production. Thus, further research into

nitrogenase is required in a multitude of areas; hence, multi-disciplinary research efforts surely present the best opportunity (and are clearly necessary) to advance nitrogenase research and understanding.

One method of improving environmental sustainability is the decentralization of key global-scale processes. To that end, nitrogenase-based technologies present opportunities for on-demand $NH_3$ production. However, much development is still required to overcome some of the limitations surrounding the utilization of nitrogenase. For example, MgATP hydrolysis presents a significant energetic requirement, and the sensitivity of nitrogenase to $O_2$ also inhibits deployment. Further, nitrogenase assembly and maturation is complex and limits in vitro utilization. Nevertheless, perhaps the most promising approach for nitrogenase deployment in new biotechnologies is presented by Liu et al., where photoelectrically generated $H_2$ feeds a $N_2$-fixing microbe for $NH_3$ production and/or biomass accumulation [144].

**Author Contributions:** W.G. and R.D.M. conceived and wrote the manuscript. All authors have read and agreed to the published version of the manuscript.

**Funding:** R.D.M. acknowledges funding from the University of Geneva.

**Conflicts of Interest:** The authors declare no conflict of interest.

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
