# Peer review of "Natural and Engineered Electron Transfer of Nitrogenase"

_chemistry, doi:10.3390/chemistry2020021_

Round 1

Reviewer 1 Report

This is a well-written review of nitrogenase, including natural and engineered electron transfer pathways. Artificial reduction pathways are considered in some detail.  These methods, which would likely be involved for any viable process of scale include electrochemical and photo-reducing strategies.  The review is comprehensive and thorough and serves as a valuable introduction to the field.

One is always left with questions about the practicality of approaches of this sort and a paragraph in the conclusion section about the requirements for commercialization would be a useful touchstone for the reader.  Is it conceivable that nitrogenase-based fixation could compete with the Haber-Bosch process?  The introduction includes a rationale for the study of new catalysts for nitrogen fixation.  What are the issues of scale that need to be solved?

Reviewer 2 Report

This is a useful and timely review of the varied approaches to bypassing the physiological reductants of nitrogenase, in order to improve efficiency and avoid the high energy cost of the physiological reaction.  The objective is the development of in vivo and in vitro technologies for improved NH3 production, in the context of improved food production.

Some minor points:

Fig 1 caption states  ‘The α-Y64, ß-Y98 and ß-F99 residues are highlighted in magenta.’ but it is not clear where they are located on the Left graphic.  Neither is it clear, at this early point in the article, why these residues are significant.  Clarification is required.

Page 3:  The discussion of mechanism adopts the reductive elimination/oxidative addition dogma.  Dance has drawn attention to the misconceptions involved in this, and points towards the valid approach involving hydrogen atoms rather than hydride ions bound to FeMo-co: see Dalton, 44, 9027 (2015) 10.1039/C5DT00771B

Mmixed V and mV units are used.

line 285:  change ‘less’ to ‘fewer’

line 324:  change ‘shares’ to ‘share’

line 349:  change ‘principals’  to  ‘principles’

line 425:  ‘production of 195’  ??
